# Probabilistic Watershed:
# Sampling all spanning forests
# for seeded segmentation and semi-supervised learning

**Enrique Fita Sanmartín,     Sebastian Damrich,     Fred A. Hamprecht**
HCI/IWR at Heidelberg University, 69115 Heidelberg, Germany
`{fita@stud, sebastian.damrich@iwr, fred.hamprecht@iwr}.uni-heidelberg.de`

## Abstract

The seeded Watershed algorithm / minimax semi-supervised learning on a graph computes a minimum spanning forest which connects every pixel / unlabeled node to a seed / labeled node. We propose instead to consider *all possible* spanning forests and calculate, for every node, the probability of sampling a forest connecting a certain seed with that node. We dub this approach "Probabilistic Watershed". Leo Grady (2006) already noted its equivalence to the Random Walker / Harmonic energy minimization. We here give a simpler proof of this equivalence and establish the computational feasibility of the Probabilistic Watershed with Kirchhoff's matrix tree theorem. Furthermore, we show a new connection between the Random Walker probabilities and the triangle inequality of the effective resistance. Finally, we derive a new and intuitive interpretation of the Power Watershed.

## 1   Introduction

Seeded segmentation in computer vision and graph-based semi-supervised machine learning are essentially the same problem. In both, a popular paradigm is the following: given many unlabeled pixels / nodes in a graph as well as a few seeds / labeled nodes, compute a distance from a given query pixel / node to all of the seeds, and assign the query to a class based on the shortest distance.

There is obviously a large selection of distances to choose from, and popular choices include: *i)* the shortest path distance (e.g. [19]), *ii)* the commute distance (e.g. [47, 46, 5, 26]) or *iii)* the bottleneck shortest path distance (e.g. [28, 12]). Thanks to its matroid property, the latter can be computed very efficiently – a greedy algorithm finds the global optimum – and is thus widely studied and used in different fields under names including widest, minimax, maximum capacity, topographic and watershed path distance. In computer vision, the corresponding algorithm known as "Watershed" is popular in seeded segmentation not only because it is so efficient [13] but also because it works well in a broad range of problems [45, 3], is well understood theoretically [17, 1], and unlike Markov Random Fields induces no shrinkage bias [4]. Even though the Watershed's optimization problem can be solved efficiently, it is combinatorial in nature. One consequence is the "winner-takes-all" characteristic of its solutions: a pixel or node is always unequivocally assigned to a single seed. Given suitable graph edge-weights, this solution is often but not always correct, see Figures 1 and 2[1].

Intrigued by the value of the Watershed to many computer vision pipelines, we have sought to entropy-regularize the combinatorial problem to make it more amenable to end-to-end learning in modern pipelines. Exploiting the equivalence of Watershed segmentations to minimum cost spanning forests, we hence set out from the following question: Is it possible to compute not just the minimum, but all (!) possible spanning forests, and to compute, in closed form, the probability that a pixel of

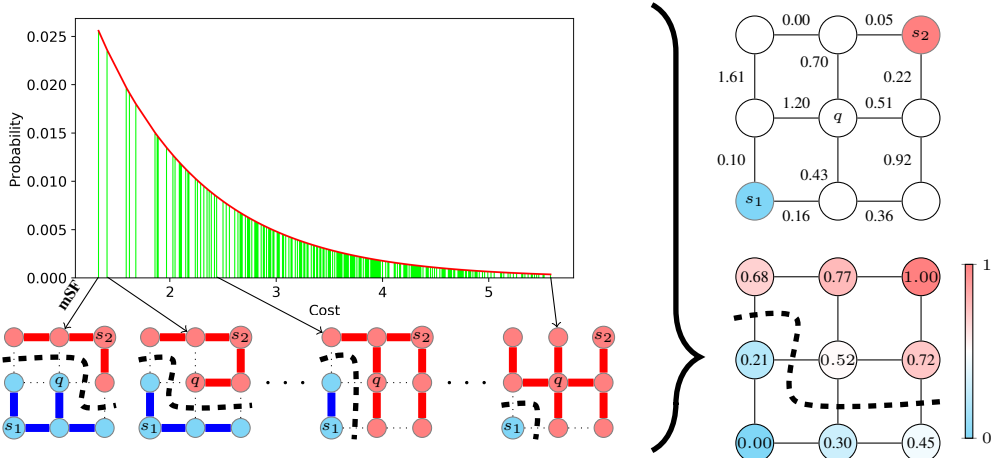

Figure 1: The Probabilistic Watershed computes the expected seed assignment of every node for a Gibbs distribution over all exponentially many spanning forests in closed-form. It thus avoids the winner-takes-all behaviour of the Watershed. (**Top right**) Graph with edge-costs and two seeds. (**Bottom left**) The minimum spanning forest (mSF) and other, higher cost forests. The Watershed selects the mSF, which assigns the query node $q$ to seed $s_1$. Other forests of low cost might however induce different segmentations. The dashed lines indicate the cut of the segmentations. For instance, the other depicted forests connect $q$ to $s_2$. (**Top left**) We therefore consider a Gibbs distribution over all spanning forests with respect to their cost (see equation (5), $\mu = 1$). Each green bar corresponds to the cost of one of the 288 possible spanning forests. (**Bottom right**) Probabilistic Watershed probabilities for assigning a node to $s_2$. Query $q$ is now assigned to $s_2$. Considering a distribution over all spanning forests gives an uncertainty measure and can yield a segmentation different from the mSF's. In contrast to the 288 forests in this toy graph, for the real-life image in Figure 2 one would have to consider at least $10^{11847}$ spanning forests separating the 13 seeds (see appendix G), a feat impossible without the matrix tree theorem.

interest is assigned to one of the seeds? More specifically, we envisaged a Gibbs distribution over the exponentially many distinct forests that span an undirected graph with edge-costs, where each forest is assigned a probability that decreases with increasing sum of the edge-costs in that forest.

If computed naively, this would be an intractable problem for all but the smallest graphs. However, we show here that a closed-form solution can be found by recurring to Kirchhoff's matrix tree theorem, and is given by the solution of the Dirichlet problem associated with commute distances [47, 46, 5, 26]. Leo Grady mentioned this connection in [26, 27] and based his argument on potential theory, using results from [8]. Our informal poll amongst experts from both computer vision and machine learning indicated that this connection has remained mostly unknown. We hence offer a completely self-contained, except for the matrix tree theorem, and hopefully simpler proof.

In this entirely conceptual work, we

- give a proof, using elementary graph constructions and building on the matrix tree theorem, that shows how to compute analytically the probability that a graph node is assigned to a particular seed in an ensemble of Gibbs distributed spanning forests (Section 3).

- establish equivalence to the algorithm known as Random Walker in computer vision [26] and as Laplacian Regularized Least Squares and under other names in transductive machine learning [47, 46, 5]. In particular, we relate, for the first time, the probability of assigning a query node to a seed to the triangle inequality of the effective resistance between seeds and query (Section 4).

- give a new interpretation of the so-called Power Watershed [15] (Section 5).

## 1.1 Related work

Watershed as a segmentation algorithm was first introduced in [6]. Since then it has been studied from different points of view [7, 16], notably as a minimum spanning forest that separates the seeds

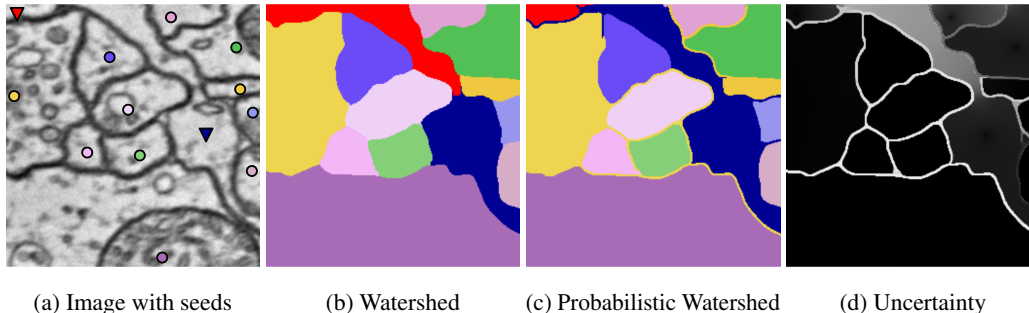

| (a) Image with seeds | (b) Watershed | (c) Probabilistic Watershed | (d) Uncertainty |

Figure 2: The Probabilistic Watershed profits from using all spanning forests instead of only the minimum cost one. (2a) Crop of a CREMI image [18] with marked seeds. (2b) and (2c) show results of Watershed and multiple seed Probabilistic Watershed (end of section 3) applied to edge-weights from [11]. (2d) shows the entropy of the label probabilities of the Probabilistic Watershed (white high, black low). The Watershed errs in an area where the Probabilistic Watershed expresses uncertainty but is correct.

[17]. The Random Walker [26, 46, 47, 5] calculates the probability that a random walker starting at a query node reaches a certain seed before the other ones. Both algorithms are related in [15] by a limit consideration termed Power Watershed algorithm. In this work, we establish a different link between the Watershed and the Random Walker. The Watershed's and Random Walker's recent combination with deep learning [45, 43, 11] also connects our Probabilistic Watershed to deep learning.

Related to our work by name though not in substance is the "Stochastic Watershed" [2, 34], which samples different instances of seeds and calculates a probability distribution over segmentation boundaries. Instead, in [38] the authors suggest sampling the edge-costs in order to define an uncertainty measure of the labeling. They show that it is NP-hard to calculate the probability that a node is assigned to a seed if the edge-costs are stochastic. We derive a closed-form formula for this probability for non-stochastic costs by sampling spanning forests. Ensemble Watersheds proposed by [12] samples part of the seeds and part of the features which determine the edge-costs. Introducing stochasticity to distance transforms makes a subsequent Watershed segmentation more robust to noise [36]. Minimum spanning trees are also applied in optimum-path forest learning, where confidence measures can be computed [21, 22]. Similar to our forest distribution, [30] considers a Gibbs distribution over shortest paths. This approach is extended to more general bags-of-paths in [24].

Entropic regularization has been used most successfully in optimal transport [20] to smooth the combinatorial optimization problem and hence afford end-to-end learning in conjunction with deep networks [35]. Similarly, we smooth the combinatorial minimum spanning forest problem by considering a Gibbs distribution over all spanning forests.

The matrix tree theorem (MTT) plays a crucial role in our theory, permitting us to measure the weight of a set of forests. The MTT is applied in machine learning [31], biology [40] and network analysis [39, 41]. The matrix forest theorem (MFT), a generalization of the MTT, is applied in [14, 37]. By means of the MFT, a distance on the graph is defined in [14]. In a similar manner as we do with the MTT, [37] is able to compute a Gibbs distribution of forests using the MFT.

Some of the theoretical results of our work are mentioned in [26, 27], where they refer to [8]. In contrast to [26], we emphasize the relation with the Watershed and develop the theory in a simpler and more direct way.

## 2 Background

### 2.1 Notation and terminology

Let $G = (V, E, w, c)$ be a graph where $V$ denotes the set of nodes, $E$ the set of edges and $w$ and $c$ are functions that assign a weight $w(e) \in \mathbb{R}_{\geq 0}$ and a cost $c(e) \in \mathbb{R}$ to each edge $e \in E$. All the graphs $G$ considered will be connected and undirected. When we speak of a multigraph, we allow for

multiple edges incident to the same two nodes but not for self-loops. We will consider simple graphs unless stated otherwise.

The Laplacian of a graph $L \in \mathbb{R}^{|V| \times |V|}$ is defined as

$$L_{uv} := \begin{cases} -w(\{u,v\}) & \text{if } u \neq v \\ \sum_{k \in V} w(\{u,k\}) & \text{if } u = v \end{cases},$$

where we consider $w(\{u,v\}) = 0$ if $\{u,v\} \notin E$. $L^+$ will denote its pseudo-inverse.

We define the weight of a graph as the product of the weights of all its edges, $w(G) = \prod_{e \in E} w(e)$. The weight of a set of graphs, $w(\{G_i\}_{i=0}^n)$ is the sum of the weights of the graphs. In a similar manner, we define the cost of a graph as the sum of the costs of all its edges, $c(G) = \sum_{e \in E} c(e)$.

The set of spanning trees of $G$ will be denoted by $\mathcal{T}$. Given a tree $t \in \mathcal{T}$ and nodes $u, v \in V$, the set of edges on the unique path between $u$ and $v$ in $t$ will be denoted by $\mathcal{P}_t(u,v)$. By $\mathcal{F}_u^v$ we denote the set of 2-trees spanning forests, i.e. spanning forests with two trees, such that $u$ and $v$ are not connected. Furthermore, if we consider a third node $q$, we define $\mathcal{F}_{u,q}^v := \mathcal{F}_u^v \cap \mathcal{F}_q^v$, i.e. all 2-trees spanning forests such that $q$ and $u$ are in one tree and $v$ belongs to the other tree. Note that the sets $\mathcal{F}_{u,q}^v (= \mathcal{F}_{q,u}^v)$ and $\mathcal{F}_{v,q}^u (= \mathcal{F}_{q,v}^u)$ form a partition of $\mathcal{F}_u^v (= \mathcal{F}_v^u)$, since $q$ must be connected either to $u$ or $v$, but not to both. In order to shorten the notation we will refer to 2-trees spanning forests simply as 2-forests.

We consider $w(e) = \exp(-\mu c(e))$, $\mu \geq 0$, as will be motivated in Section 3.1 by the definition of a Gibbs distribution over the 2-forests in $\mathcal{F}_u^v$. Thus, a low edge-cost corresponds to a large edge-weight, and a minimum edge-cost spanning forest (mSF) is equivalent to a maximum edge-weight spanning forest (MSF).

## 2.2 Seeded Watershed as minimum cost spanning forest computation

Let $G = (V, E, c)$ be a graph and $c(e)$ be the cost of edge $e$. The lower the cost, the higher the affinity between the nodes incident to $e$. Given different seeds, a forest in the graph defines a segmentation over the nodes as long as each component contains a different seed. The cost of a forest, $c(f)$, is equal to the sum of the costs of its edges. The Watershed algorithm calculates a minimum cost spanning forest, mSF, (or maximum weight, MSF) such that the seeds belong to different components [17].

## 2.3 Matrix tree theorem

In our approach we want to take all possible 2-forests in $\mathcal{F}_u^v$ into account. The probability of a node label will be measured by the cumulative weight of the 2-forests connecting the node to a seed of that label. To compute the weight of a set of 2-forests we will use the matrix tree theorem (MTT) which can be found e.g. in chapter 4 of [42] (see Appendix A) and has its roots in [29].

**Theorem 2.1** (MTT). For any edge-weighted multigraph $G$ the sum of the weights of the spanning trees of $G$, $w(\mathcal{T})$, is equal to

$$w(\mathcal{T}) := \sum_{t \in \mathcal{T}} w(t) = \sum_{t \in \mathcal{T}} \prod_{e \in E_t} w(e) = \frac{1}{|V|} \det\left(L + \frac{1}{|V|} \mathbb{1}\mathbb{1}^\top\right) = \det(L^{[v]}),$$

where $\mathbb{1}$ is a column vector of 1's. $L^{[v]}$ is the matrix obtained from $L$ after removing the row and column corresponding to an arbitrary but fixed node $v$.

This theorem considers trees instead of 2-forests. The key idea to obtain an expression for $w(\mathcal{F}_u^v)$ by means of the MTT is that any 2-forest $f \in \mathcal{F}_u^v$ can be transformed into a tree by adding an artificial edge $\bar{e} = \{u, v\}$ which connects the two components of $f$ (as done in section 9 of [8] or in the original work of Kirchhoff [29]). We obtain the following lemma, which is proven in Appendix A.

*Lemma* 2.2. Let $G = (V, E, w)$ be an undirected edge-weighted connected graph and $u, v \in V$ arbitrary vertices.

a) Let $\ell_{ij}^+$ denote the entry $ij$ of the pseudo-inverse of the Laplacian of $G$, $L^+$. Then we get

$$w(\mathcal{F}_u^v) = w(\mathcal{T})\left(\ell_{uu}^+ + \ell_{vv}^+ - 2\ell_{uv}^+\right). \tag{1}$$

b) Let $\ell_{ij}^{-1,[r]}$ denote the entry $ij$ of the inverse of the matrix $L^{[r]}$ (the Laplacian $L$ after removing the row and the column corresponding to node $r$), then

$$w(\mathcal{F}_u^v) = \begin{cases} w(\mathcal{T})\left(\ell_{uu}^{-1,[r]} + \ell_{vv}^{-1,[r]} - 2\ell_{uv}^{-1,[r]}\right) & \text{if } r \neq u, v \\ w(\mathcal{T})\ell_{uu}^{-1,[v]} & \text{if } r = v \text{ and } u \neq v \\ w(\mathcal{T})\ell_{vv}^{-1,[u]} & \text{if } r = u \text{ and } u \neq v. \end{cases} \qquad (2)$$

## 2.4 Effective resistance

In electrical network theory, the circuits are also interpreted as graphs, where the weights of the edges are defined by the reciprocal of the resistances of the circuit. The effective resistance between two nodes $u$ and $v$ can be defined as $r_{uv}^{\text{eff}} := (\nu_u - \nu_v)/I$ where $\nu_u$ is the potential at node $u$ and $I$ is the current flowing into the network. Other equivalent expressions for the effective resistance [25] in terms of the matrices $L^+$ and $L^{[r]}$, as defined in Lemma 2.2, are

$$r_{uv}^{\text{eff}} = \ell_{uu}^+ + \ell_{vv}^+ - 2\ell_{uv}^+ = \begin{cases} \left(\ell_{uu}^{-1,[r]} + \ell_{vv}^{-1,[r]} - 2\ell_{uv}^{-1,[r]}\right) & \text{if } r \neq u, v \\ \ell_{uu}^{-1,[v]} & \text{if } r = v \text{ and } u \neq v \\ \ell_{vv}^{-1,[u]} & \text{if } r = u \text{ and } u \neq v. \end{cases} \qquad (3)$$

We observe that the expressions in Lemma 2.2 and in equation (3) are proportional. We will develop this relation further in Section 3.2. An important property of the effective resistance is that it defines a metric over the nodes of a graph ([23] Section 2.5.2).

## 3 Probabilistic Watershed

Instead of computing the mSF, as in the Watershed algorithm, we take into account all the 2-forests that separate two seeds $s_1$ and $s_2$ in two trees according to their costs. Since each 2-forest assigns a query node to exactly one of the two seeds, we calculate the probability of sampling a 2-forest that connects the seed with the query node. Moreover, this provides an uncertainty measure of the assigned label. We call this approach to semi-supervised learning "Probabilistic Watershed".

### 3.1 Probability of connecting two nodes in an ensemble of 2-forests

In Section 2.1, we defined the cost of a forest as the cumulative cost of its edges. We assume that the 2-forests $f \in \mathcal{F}_{s_1}^{s_2}$ follow a probability distribution that minimizes the expected cost of a 2-forest among all distributions of given entropy $J$. Formally, the 2-forests are sampled from the distribution which minimizes

$$\min_P \sum_{f \in \mathcal{F}_{s_1}^{s_2}} P(f)c(f), \quad \text{s.t.} \quad \sum_{f \in \mathcal{F}_{s_1}^{s_2}} P(f) = 1 \text{ and } \mathcal{H}(P) = J, \qquad (4)$$

where $\mathcal{H}(P)$ is the entropy of $P$. The lower the entropy, the more probability mass is given to the 2-forests of lowest cost. The minimizing distribution is the Gibbs distribution (e.g. [44] 3.2):

$$P(f) = \frac{\exp(-\mu c(f))}{\sum_{f' \in \mathcal{F}_{s_1}^{s_2}} \exp(-\mu c(f'))} = \frac{\prod_{e \in E_f} \exp(-\mu c(e))}{\sum_{f' \in \mathcal{F}_{s_1}^{s_2}} \prod_{e \in E_{f'}} \exp(-\mu c(e))} = \frac{w(f)}{\sum_{f' \in \mathcal{F}_{s_1}^{s_2}} w(f')}, \qquad (5)$$

where $\mu$ implicitly determines the entropy. A higher $\mu$ implies a lower entropy (see Section 5 and Figure 1 in the appendix). According to (5), an appropriate choice for the edge-weights is $w(e) = \exp(-\mu c(e))$. The main definition of the paper is:

**Definition 3.1 (Probabilities of the Probabilistic Watershed).** Given two seeds $s_1$ and $s_2$ and a query node $q$, we define the Probabilistic Watershed's probability that $q$ and $s_1$ have the same label as the probability of sampling a 2-forest that connects $s_1$ and $q$, while separating the seeds:

$$P(q \sim s_1) := \sum_{f \in \mathcal{F}_{s_1,q}^{s_2}} P(f) = \sum_{f \in \mathcal{F}_{s_1,q}^{s_2}} w(f) \Big/ \sum_{f' \in \mathcal{F}_{s_1}^{s_2}} w(f') = w\left(\mathcal{F}_{s_1,q}^{s_2}\right) \Big/ w\left(\mathcal{F}_{s_1}^{s_2}\right). \qquad (6)$$

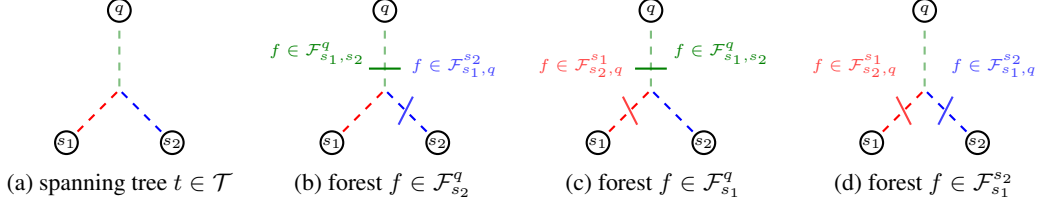

(a) spanning tree $t \in \mathcal{T}$    (b) forest $f \in \mathcal{F}_{s_1}^{s_2}$    (c) forest $f \in \mathcal{F}_{s_1}^q$    (d) forest $f \in \mathcal{F}_{s_1}^{s_2}$

Figure 3: Amongst all spanning forests that isolate seed $s_1$ from $s_2$, we want to identify the fraction of forests connecting $s_1$ and $q$ (Definition 3.1). The dashed lines represent all spanning trees. Either cut in (3b) yields a forest separating $q$ from $s_2$. The blue ones are of interest to us. Diagrams (3b) - (3d) correspond to the three equations in the linear system (7), which can be solved for $w(\mathcal{F}_{s_1,q}^{s_2})$.

The Watershed algorithm computes a minimum cost 2-forest, which is the most likely 2-forest according to (5), and segments the nodes by their connection to seeds in the minimum cost spanning 2-forest. However, it does not indicate which label assignments were ambiguous, for instance due to the existence of other low - but not minimum - cost 2-forests. This makes it a brittle "winner-takes-all" approach. In contrast, the Probabilistic Watershed takes all spanning 2-forests into account according to their cost (see Figure 1). The resulting assignment probability of each node provides an uncertainty measure. Assigning each node to the seed for which it has the highest probability can yield a segmentation different from the Watershed's.

## 3.2    Computing the probability of a query being connected to a seed

In the previous subsection, we defined the probability of a node being assigned to a seed via a Gibbs distribution over all exponentially many 2-forests. Here, we show that it can be computed analytically using only elementary graph constructions and the MTT (Theorem 2.1). In Lemma 2.2 we have stated how to calculate $w(\mathcal{F}_u^v)$ for any $u, v \in V$. Applying this to $\mathcal{F}_{s_1}^{s_2}$, $\mathcal{F}_{s_1}^q$ and $\mathcal{F}_{s_2}^q$ we can compute $w(\mathcal{F}_{s_1,q}^{s_2})$ and $w(\mathcal{F}_{s_2,q}^{s_1})$ by means of a linear system.

$\mathcal{F}_{u,q}^v$ and $\mathcal{F}_{v,q}^u$ form a partition of $\mathcal{F}_u^v$ for any mutually distinct nodes $u, v, q$ as mentioned in Section 2.1. Thus, we obtain the linear system of three equations in three unknowns:

$$\begin{array}{rcl} w(\mathcal{F}_{s_1,q}^{s_2}) + w(\mathcal{F}_{s_1,s_2}^q) & = & w(\mathcal{F}_{s_2}^q) \\ w(\mathcal{F}_{s_1,s_2}^q) + w(\mathcal{F}_{s_2,q}^{s_1}) & = & w(\mathcal{F}_{s_1}^q) \\ w(\mathcal{F}_{s_1,q}^{s_2}) + w(\mathcal{F}_{s_2,q}^{s_1}) & = & w(\mathcal{F}_{s_1}^{s_2}). \end{array} \qquad (7)$$

In this paragraph, we describe an alternative way of deriving (7) by relating spanning 2-forests to spanning trees before we solve it in (8). This is similar to our use of the MTT for counting spanning 2-forests instead of trees in Lemma A.4 (see Appendix A). Let $t$ be a spanning tree of $G$. To create a 2-forest $f \in \mathcal{F}_{s_1}^{s_2}$ from $t$ we need to remove an edge $e$ in the path from $s_1$ to $s_2$, that is $e \in \mathcal{P}_t(s_1, s_2)$. This edge $e$ must be either in $\mathcal{P}_t(q, s_1) \cap \mathcal{P}_t(s_1, s_2)$ or $\mathcal{P}_t(q, s_2) \cap \mathcal{P}_t(s_1, s_2)$ (shown in red and blue respectively in Figure 3d), as the union of $P_t(s_1, q)$ and $P_t(q, s_2)$ contains $P_t(s_1, s_2)$ and removing $e$ from $t$ cannot pairwise separate $q$, $s_1$ and $s_2$. If we remove an edge from $\mathcal{P}_t(q, s_2) \cap \mathcal{P}_t(s_1, s_2)$, we get $f \in \mathcal{F}_{s_1,q}^{s_2}$ since we are disconnecting $s_2$ from $q$, otherwise $f \in \mathcal{F}_{s_2,q}^{s_1}$. Analogously, we obtain a 2-forest in $\mathcal{F}_{s_1}^q$ or $\mathcal{F}_{s_2}^q$ if we remove an edge $e$ from $\mathcal{P}_t(s_1, q)$ or $\mathcal{P}_t(s_2, q)$ respectively (see Figure 3). When applied to all spanning trees, we obtain the system (7).

Solving the linear system (7) we obtain [2]

$$w\left(\mathcal{F}_{s_1,q}^{s_2}\right) = \left(w(\mathcal{F}_{s_2}^q) + w(\mathcal{F}_{s_1}^{s_2}) - w(\mathcal{F}_{s_1}^q)\right)/2. \qquad (8)$$

In consequence of equation (8) and Definition 3.1 we get the following theorem:

**Theorem 3.1.** The probability that $q$ has the same label as seed $s_1$ is

$$P(q \sim s_1) = \left(w(\mathcal{F}_{s_2}^q) + w(\mathcal{F}_{s_1}^{s_2}) - w(\mathcal{F}_{s_1}^q)\right)\big/\left(2w(\mathcal{F}_{s_1}^{s_2})\right).$$

Theorem 3.1 expresses $P(q \sim s_1)$ in terms of weights of 2-forests, which we can compute with Lemma 2.2, which is based on the MTT. We use this expression to relate $P(q \sim s_1)$ to the effective resistance. As a result of Lemma 2.2 and equation (3), for any nodes $u, v \in V$ we have

$$r_{uv}^{\mathrm{eff}} = w\left(\mathcal{F}_u^v\right) / w(\mathcal{T}). \tag{9}$$

This relation has already been proven in [8] (Proposition 17.1) but in terms of the effective conductance (the inverse of the effective resistance). Due to $r_{uv}^{\mathrm{eff}}$ being a metric, $w\left(\mathcal{F}_u^v\right)$ also defines a metric over the nodes of the graph. Combining (9) with Theorem 3.1, we have that the probability of $q$ having seed $s_1$'s label is

$$P(q \sim s_1) = \left(r_{s_2 q}^{\mathrm{eff}} + r_{s_2 s_1}^{\mathrm{eff}} - r_{s_1 q}^{\mathrm{eff}}\right) / \left(2 r_{s_1 s_2}^{\mathrm{eff}}\right) \tag{10}$$

The probability is proportional to the gap in the triangle inequality $r_{s_1 q}^{\mathrm{eff}} \leq r_{s_1 s_2}^{\mathrm{eff}} + r_{s_2 q}^{\mathrm{eff}}$. It will be shown in Section 4 that the probability defined in Definition 3.1 is equal to the probability given by the Random Walker [26]. Equation (10) gives an interpretation of this probability, which is new to the best of our knowledge. We can see that the greater the gap in the triangle inequality, the greater is the probability. Further, we get $P(q \sim s_1) \geq P(q \sim s_2) \iff r_{s_1 q}^{\mathrm{eff}} \leq r_{s_2 q}^{\mathrm{eff}}$. This relation has already been pointed out in [26] (section IV.B) in terms of the effective conductance between two nodes, but not as explicitly as in (10). We note that any metric distance on the nodes of a graph, e.g. the ones mentioned in the introduction, can define an assignment probability along the lines of equation (10).

Our discussion was constrained to the case of two seeds only to ease our explanation. We can reduce the case of multiple seeds per label to the two seed case by merging all nodes seeded with the same label. Similarly, the case of more than two labels can be reduced to the two label scenario by using a one versus all strategy: We choose one label and merge the seeds of other labels into one unique seed. In both cases we might introduce multiple edges between node pairs. While having formulated our arguments for simple graphs, they are also valid for multigraphs (see Appendix A).

## 4 Connection between the Probabilistic Watershed and the Random Walker

In this section we will show that the Random Walker of [26] is equivalent to our Probabilistic Watershed, both computationally and in terms of the resulting label probabilities.

**Theorem 4.1.** The probability $x_q^{s_1}$ that a random walker as defined in [26] starting at node $q$ reaches $s_1$ first before reaching $s_2$ is equal to the Probabilistic Watershed probability defined in Definition 3.1:

$$x_q^{s_1} = P(q \sim s_1).$$

This equivalence, which we prove in Appendix B, was pointed out by Leo Grady in [26] section IV.B but with a different approach. Grady relied on results from [8], where potential theory is used. There it is shown that $x_q^{s_1} = w(\mathcal{F}_{s_1,q}^{s_2}) / \left(r_{s_1 s_2}^{\mathrm{eff}} w(\mathcal{T})\right)$. From this formula we get Theorem 4.1 by using equation (9):

$$x_q^{s_1} = w(\mathcal{F}_{s_1,q}^{s_2}) / \left(r_{s_1 s_2}^{\mathrm{eff}} w(\mathcal{T})\right) = w\left(\mathcal{F}_{s_1,q}^{s_2}\right) / w\left(\mathcal{F}_{s_1}^{s_2}\right) = P(q \sim s_1).$$

We have proven the same statement with elementary arguments and without the main theory of [8]. Through the use of the MTT, we have shown that the forest-sampling point of view is computationally equivalent to the in practice very useful Random Walker (see [47, 26], and recently [43, 10, 11, 32, 9]), making our method just as potent. We thus refrained from adding further experiments and instead include a new interpretation of the Power Watershed within our framework.

## 5 Power Watershed counts minimum cost spanning forests

The objective of this section is to recall the Power Watershed [15] (see Appendix C for a summary) and develop a new understanding of its nature. Power Watershed is a limit over the Random Walker and thus over the equivalent Probabilistic Watershed. The latter's idea of measuring the weight of a set of 2-forests carries over nicely to the Power Watershed, where, as a limit, only the maximum weight / minimum cost spanning forests are considered. This section details the connection.

Let $G = (V, E, w, c)$ and $s_1, s_2 \in V$ be as before. In [15] the following objective function is proposed:

$$\arg\min_x \sum_{e=\{u,v\} \in E} (w(e))^\alpha \left(|x_u - x_v|\right)^\beta, \text{ s.t. } x_{s_1} = 1, \ x_{s_2} = 0. \tag{11}$$

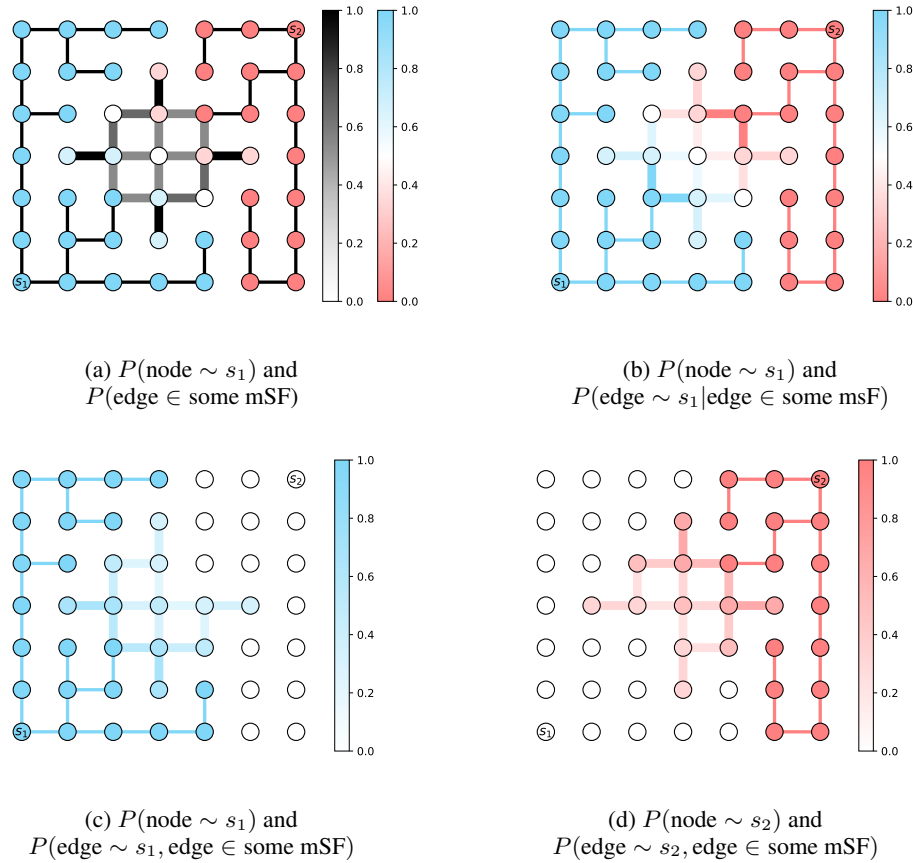

(a) $P(\text{node} \sim s_1)$ and
$P(\text{edge} \in \text{some mSF})$

(b) $P(\text{node} \sim s_1)$ and
$P(\text{edge} \sim s_1 | \text{edge} \in \text{some mSF})$

(c) $P(\text{node} \sim s_1)$ and
$P(\text{edge} \sim s_1, \text{edge} \in \text{some mSF})$

(d) $P(\text{node} \sim s_2)$ and
$P(\text{edge} \sim s_2, \text{edge} \in \text{some mSF})$

Figure 4: Power Watershed result on a grid graph with seeds $s_1$, $s_2$ and with random edge-costs outside a plateau of edges with the same cost (wide edges). By the results in Theorem 5.1, the Power Watershed counts mSFs. This is illustrated both with the node- and edge-colors. (4a-4d) The nodes are colored by their probability of belonging to seed $s_1$ ($s_2$), i.e. by the share of mSFs that connect a given node to $s_1$ ($s_2$). (4a) The edge-color indicates the share of mSFs in which the edge is present. (4b) The edge-color indicates the share of mSFs in which the edge is connected to seed $s_1$ among the mSFs that contain the edge. (4c - 4d) The edge-color indicates the share of mSFs in which the edge is connected to $s_1$ or $s_2$, respectively, among all mSFs. See Appendix F for a more detailed explanation.

For $\alpha = 1$ and $\beta = 2$ it gives the Random Walker's objective function. The Power Watershed considers the limit case when $\alpha \to \infty$ and $\beta$ remains finite.

In section 3.1 we defined the weight of an edge $e$ as $w(e) = \exp(-\mu c(e))$, where $c(e)$ was the edge-cost and $\mu$ implicitly determined the entropy of the 2-forest distribution. By raising the weight of the edges to $\alpha$ we obtain $\left(w(e)\right)^\alpha = \exp(-\mu \alpha c(e)) = \exp(-\mu_\alpha c(e))$, where $\mu_\alpha := \mu \alpha$. Therefore, we can absorb $\alpha$ into $\mu$. When $\alpha \to \infty$ (and therefore $\mu_\alpha \to \infty$) the distribution will have a lowest entropy. As a consequence only the mSFs / MSFs are considered in the Power Watershed:

**Theorem 5.1.** Given two seeds $s_1$ and $s_2$, let us denote the potential of node $q$ being assigned to seed $s_1$ by the Power Watershed with $\beta = 2$ as $x_q^{\text{PW}}$. Let further $w_{\max}$ be $\max_{f \in \mathcal{F}_{s_1}^{s_2}} w(f)$. Then

$$x_q^{\text{PW}} = \frac{\left|\{f \in \mathcal{F}_{s_1,q}^{s_2} \; : \; w(f) = w_{\max}\}\right|}{\left|\{f \in \mathcal{F}_{s_1}^{s_2} \; : \; w(f) = w_{\max}\}\right|}.$$

Theorem 5.1, which we prove in Appendix D, interprets the Power Watershed potentials as a ratio of 2-forests similar to the Probabilistic Watershed. But instead of all 2-forests the Power Watershed only considers minimum cost 2-forests (equivalently maximum weight 2-forests) as they are the only ones that matter after taking the limit $\mu \to \infty$ (or $\alpha \to \infty$). In other words, the Power Watershed counts

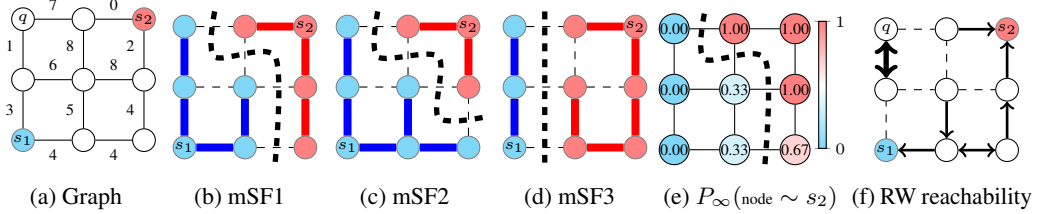

|  (a) Graph | (b) mSF1 | (c) mSF2 | (d) mSF3 | (e) $P_\infty(\text{node} \sim s_2)$ | (f) RW reachability |

Figure 5: Forest-interpretation of Power Watershed. (5a) Graph with edge-costs and its mSFs in ((5b)-(5d)). (5e) Power Watershed probabilities for assigning a node to $s_2$. The Power Watershed computes the ratio between the mSFs connecting a node to $s_2$ and all possible mSFs. The dashed lines indicate the segmentation's cut. (5f) indicates the allowed Random Walker transitions when $\mu \to \infty$ with headed arrows. The Random Walker interpretation of the Power Watershed breaks down in the limit case since a Random Walker starting at node $q$ does not reach any seed, but oscillates along the bold arrow.

by how many seed separating mSFs a node is connected to a seed (see Figure 5). Note, that there can be more than one mSF when the edge-costs are not unique. In Figure 4 we show the probability of an edge being part of a mSF (see Appendix F for a more exhaustive explanation). In addition, it is worth recalling that the cut given by the Power Watershed segmentation is a mSF-cut (Property 2 of [15]).

The Random Walker interpretation can break down in the limit case of the Power Watershed. After taking the power of the edge-weights to infinity, at any node a Random Walker would move along an incident edge with maximum weight / minimum cost. So, in the limit case a Random Walker could get stuck at the edges, $e = \{u, v\}$, which minimize the cost among all the edges incident to $u$ or $v$. In this case the Random Walker will not necessarily reach any seed (see Figure 5f). In contrast, the forest-counting interpretation carries over nicely to the limit case.

The Probabilistic Watershed with a Gibbs distribution over 2-forests of minimal (maximal) entropy, $\mu = \infty$ ($\mu = 0$), corresponds to the Power Watershed (only considers the graph's topology). The effect of $\mu$ is illustrated on a a toy graph in Figure 1 of the appendix. One could perform grid search to identify interesting intermediate values of $\mu$. Alternatively, $\mu$ can be learned, alongside the edge-costs, by back-propagation [11] or by a first-order approximation thereof [43].

## 6  Discussion

In this work, we provided new understanding of well-known seeded segmentation algorithms.

We have presented a tractable way of computing the expected label assignment of each node by a Gibbs distribution over all the seed separating spanning forests of a graph (Definition 3.1). Using the MTT we showed that this is computationally and by result equivalent to the Random Walker [26]. Our approach has been developed without using potential theory (in contrast to [8]).

These facts have provided us with a novel understanding of the Random Walker (Probabilistic Watershed) probabilities: They are proportional to the gap produced by the triangle inequality of the effective resistance between the seeds and the query node.

Finally, we have proposed a new interpretation of the Power Watershed potentials for $\beta = 2$ and $\alpha \to \infty$: They are given as the probabilities of the Probabilistic Watershed when the latter is restricted to mSFs instead of all spanning forests.

A mSF can also be seen as a union of minimax paths between the vertices [33]. Recently, [12] showed that the Power Watershed assigns a query node $q$ to the seed to which the minimax path from $q$ has the lowest maximum edge cost. In future work, we hope to extend this path-related point of view to an intuitive understanding of the Power Watershed.

We are currently working on an extension of the Probabilistic Watershed framework to directed graphs, by means of the generalization of the MTT to directed graphs [42]. Here, one samples directed spanning forests with the seeds as sinks to segment the unlabelled nodes. This might lead to a new practical algorithm for semi-supervised learning on directed graphs such as social / citation or Web networks and could be related to directed random walks.

## Acknowledgements

The authors would like to thank Prof. Marco Saerens for his profound and constructive comments as well as the anonymous reviewers for their helpful remarks. We would like to express our gratitude to Lorenzo Cerrone, who also shared the edge weights of [11], and Laurent Najman for the useful discussions about the Random Walker and Power Watershed algorithms, respectively. We also acknowledge partial financial support of the DFG under grant No. DFG HA-4364 8-1.

## Footnotes

[1]which were produced with the code at `https://github.com/hci-unihd/Probabilistic_Watershed`

[2]Section IV.B of [26] states $w(\mathcal{F}_{u,q}^v) = w(\mathcal{F}_u^v) - w(\mathcal{F}_v^q)$ for any $u, v, q \in V$ but that formula is incorrect. For instance, it does not hold for the complete graph with nodes $\{u, v, q\}$ and with $w(e) = 1$ for all edges $e$, since $w(\mathcal{F}_{u,q}^v) = 1 \neq 0 = 2 - 2 = w(\mathcal{F}_u^v) - w(\mathcal{F}_v^q)$.

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
