[Supplementary Material]

# Supplementary Material for NeurIPS 2019
# Probabilistic Watershed:
# Sampling all spanning forests
# for seeded segmentation and semi-supervised learning

**Enrique Fita Sanmartín,    Sebastian Damrich,    Fred A. Hamprecht**
HCI/IWR at University of Heidelberg, 69115 Heidelberg, Germany
{fita@stud, sebastian.damrich@iwr, fred.hamprecht@iwr}.uni-heidelberg.de

## A    Calculus of $w\left(\mathcal{F}_u^v\right)$

In this appendix we will focus on the calculation of $w(\mathcal{F}_u^v)$ for any nodes $u, v \in V$. We will prove Lemma 2.2 (here denoted Lemma $A.5$). In order to demonstrate it we will use the matrix tree theorem (MTT) and some previous results.

To deduce some results we will use multigraphs. The Laplacian of a multigraph is slightly different from the Laplacian of a simple graph. We introduce the following definition.

**Definition A.1** (**Laplacian of a multigraph**)**.** Let $G$ be a multigraph. The Laplacian $L_G$ has the following formula

$$\left(L_G\right)_{uv} := \begin{cases} \sum_{\bar{e} \in E^{(u,v)}} -w_G\left(\bar{e}\right) & \text{if } u \neq v \\ \sum_{\substack{k \in V \\ k \neq u}} \sum_{\bar{e} \in E^{(u,k)}} w_G\left(\bar{e}\right) & \text{if } u = v \end{cases},$$

where $E^{(u,v)} \subset E_G$ is the subset of edges incident to $u$ and $v$. If there are no edges incident to $u$ and $v$ the sum is considered equal to $0$.

Since the MTT is crucial in our theory we recall it.

**Theorem A.1** (**Matrix tree theorem (MTT)**)**.** For any weighted multigraph $G$ the sum of the weights of the spanning trees of $G$, $w(\mathcal{T}) := \sum_{t \in \mathcal{T}} \prod_{e \in E_t} w(e)$, is equal to

$$w(\mathcal{T}) = \det(L^{[v]}) = \frac{1}{|V|} \lambda_2 \cdots \lambda_{|V|} = \frac{1}{|V|} \det(L + \frac{1}{|V|} \mathbb{1}\mathbb{1}^\top),$$

where $L^{[v]}$ is the matrix obtained from the Laplacian, $L$, after removing the row and column corresponding to an arbitrary but fixed node $v$, $\{\lambda_i\}_{i \geq 1}$ are the eigenvalues of L with $\lambda_1 = 0$ and $\mathbb{1}$ the corresponding eigenvector, the column vector of 1's.

*Proof.* We will only prove the third equality. The first equality can be found in [11]. The second equality can be found in chapter 1 of [2] for non-weighted graphs, but the reasoning of the proof is equivalent for weighted graphs. The proof of the third equality is Theorem 1.6 in [5], but since it has some typos we preferred to prove it by means of the second equality.

In order to demonstrate the third equality we will show that

$$\lambda_2 \cdots \lambda_{|V|} = \det(L + \frac{1}{|V|} \mathbb{1}\mathbb{1}^\top). \tag{1}$$

Let $\tilde{\lambda}_i$ for $i = 1, \ldots, |V|$ be the eigenvalues of $L + \frac{1}{|V|} \mathbb{1}\mathbb{1}^\top$. Since the determinant of a matrix is the product of its eigenvalues we obtain

$$\det(L + \frac{1}{|V|} \mathbb{1}\mathbb{1}^\top) = \tilde{\lambda}_1 \cdots \tilde{\lambda}_{|V|}.$$

We will show that one of the $\tilde{\lambda}_i$'s, say $\tilde{\lambda}_1$, is one and that $\{\tilde{\lambda}_2, \ldots, \tilde{\lambda}_{|V|}\} = \{\lambda_2, \ldots, \lambda_{|V|}\}$ which establishes equation (1).

The first eigenvalue of the Laplacian $L$ is $\lambda_1 = 0$ whose eigenvector is $\mathbb{1}$, since the elements of every row of $L$ sum to 1. We prove now that $\mathbb{1}$ is an eigenvector of $L + \frac{1}{|V|} \mathbb{1}\mathbb{1}^\top$ with eigenvalue equal to 1.

$$(L + \frac{1}{|V|} \mathbb{1}\mathbb{1}^\top)\mathbb{1} = \underbrace{L\mathbb{1}}_{=0} + \frac{1}{|V|} \mathbb{1}\underbrace{\mathbb{1}^\top \mathbb{1}}_{=|V|} = \mathbb{1}$$

Therefore, we get $\tilde{\lambda}_1 = 1$. Since $L + \frac{1}{|V|} \mathbb{1}\mathbb{1}^\top$ is symmetric, we can find an orthogonal basis of eigenvectors of $L + \frac{1}{|V|} \mathbb{1}\mathbb{1}^\top$ containing $\mathbb{1}$. Let $x_i$ be an element of that basis associated with $\lambda_i$ for $i \geq 2$. By the orthogonality of $\mathbb{1}$ and $x_i$, we get

$$(L + \frac{1}{|V|} \mathbb{1}\mathbb{1}^\top)x_i = Lx_i + \frac{1}{|V|} \mathbb{1}\underbrace{\mathbb{1}^\top x_i}_{=0} = \lambda_i x_i$$

Therefore $\lambda_i$ for $i \geq 2$ is also an eigenvalue of $L + \frac{1}{|V|} \mathbb{1}\mathbb{1}^\top$ and the theorem is proven. $\qquad\square$

*Lemma* A.2 (Determinant Lemma). Given an invertible matrix $A \in \mathbb{R}^{m \times m}$ and $u, v \in \mathbb{R}^m$ then:

$$\det(A + uv^\top) = \det(A)(1 + v^\top A^{-1} u)$$

*Proof.* See [7]. $\qquad\square$

*Lemma* A.3. Let $L^+$ be the pseudo-inverse of the Laplacian, then

$$L^+ = \left(L + \frac{\mathbb{1}\mathbb{1}^\top}{|V|}\right)^{-1} - \frac{\mathbb{1}\mathbb{1}^\top}{|V|},$$

where $\mathbb{1}$ is the column vector of 1s.

*Proof.* On page 48 of [4] the same argument as in the proof of Theorem $A.1$ is applied. $\left(L + \frac{\mathbb{1}\mathbb{1}^\top}{|V|}\right)^{-1}$ and $L^+$ have the same eigenvectors and eigenvalues except for $\tilde{\lambda}_1 = 1$ whose corresponding eigenvalue of $L^+$ is $\lambda_1 = 0$. By subtracting $\frac{\mathbb{1}\mathbb{1}^\top}{|V|}$ from $\left(L + \frac{\mathbb{1}\mathbb{1}^\top}{|V|}\right)^{-1}$ the eigenvalue $\tilde{\lambda}_1$ is modified and becomes 0. Hence both matrices are equal since they have the same spectral decomposition . See chapter 10 in [9] for more details. $\qquad\square$

Before proving the next result we need to introduce some notation. Let $\bar{e} = \{u, v\} \subset V$ be an edge not necessarily included in $E$. Let $G_{\bar{e}}$ be the graph formed from $G$ after adding the edge $\bar{e}$, i.e. $G_{\bar{e}} = (V, E \sqcup \{\bar{e}\})$, where $\sqcup$ denotes the disjoint union[1]. The use of the disjoint union permits distinguishing between the added edge and the ones originally present in the graph. Therefore $G_{\bar{e}}$ may be a multigraph. The following lemma explains why we need to consider $G_{\bar{e}}$ as a multigraph.

*Lemma* A.4. Let $G = (V_G, E_G, w_G)$ be a weighted graph and consider $G_{\bar{e}} = (V_G, E_G \sqcup \{\bar{e}\}, w)$ for some $\bar{e} = \{u, v\} \subset V$, where $w(e) = w_G(e)$ for all $e \in E_G$ and $w(\bar{e})$ an arbitrary positive number. We will omit the subscript of $w_G$ without risk of confusion. Then

$$w\left(\mathcal{F}_u^v\right) = \frac{w(\mathcal{T}_{G_{\bar{e}}}) - w(\mathcal{T}_G)}{w(\bar{e})},$$

where $\mathcal{T}_{G_{\bar{e}}}$ and $\mathcal{T}_G$ denote the set of spanning trees of $G_{\bar{e}}$ and $G$ respectively.

*Proof.* The key idea of the proof is the fact that $\mathcal{T}_{G_{\bar{e}}}$ can be partitioned in the set of trees that do not contain the edge $\bar{e}$ (which is equal to $\mathcal{T}_G$ since $G$ did not contain $\bar{e}$) and the ones that do contain the edge $\bar{e}$. Let us denote the second set $\mathcal{T}^{\bar{e}} := \{t \in \mathcal{T}_{G_{\bar{e}}} : \bar{e} \in E_t\}$. Recall that $\bar{e}$ is considered as a special edge even if there was already an edge $e = \{u, v\}$ contained in the graph $G$. If $e$ and $\bar{e}$ where considered as the same edge, then the set of trees not containing $\bar{e}$ would not be equal to $\mathcal{T}_G$, since $e$ belongs to some trees in $\mathcal{T}_G$. Therefore we need to consider $G_{\bar{e}}$ as a multigraph.

Note that there is a bijection between $\mathcal{T}^{\bar{e}}$ and $\mathcal{F}_u^v$ since any tree $t \in \mathcal{T}^{\bar{e}}$ forms a 2-forest $f \in \mathcal{F}_u^v$ after removing $\bar{e}$ from $t$, and vice versa, any $f \in \mathcal{F}_u^v$ forms a tree in $t \in \mathcal{T}^{\bar{e}}$ after adding $\bar{e}$ (see p.652 in [1]). Moreover, $w(\bar{e}) \cdot w(f) = w(t)$ since the only edge present in $t$ but not in $f$ is $\bar{e}$. Therefore, we obtain

$$\begin{aligned} w(\mathcal{T}_{G_{\bar{e}}}) &= \sum_{t \in \mathcal{T}_{G_{\bar{e}}}} w(t) = \sum_{t \in \mathcal{T}^{\bar{e}}} w(t) + \sum_{t \in \mathcal{T}_G} w(t) = w(\bar{e}) \sum_{t \in \mathcal{T}^{\bar{e}}} \frac{w(t)}{w(\bar{e})} + w(\mathcal{T}_G) \\ &= w(\bar{e}) \sum_{f \in \mathcal{F}_u^v} w(f) + w(\mathcal{T}_G) = w(\bar{e}) w(\mathcal{F}_u^v) + w(\mathcal{T}_G). \end{aligned} \tag{2}$$

Isolating $w(\mathcal{F}_u^v)$ we get the desired result. $\qquad\square$

*Lemma A.5* (**Lemma 2.2**). Let $G = (V, E, w)$ be an undirected edge-weighted connected graph and $u, v \in V$ arbitrary vertices.

(a) Let $\ell_{ij}^+$ denote the entry $ij$ of the pseudo-inverse of the Laplacian of $G$, $L_G^+$, then

$$w(\mathcal{F}_u^v) = w(\mathcal{T}) \left( \ell_{uu}^+ + \ell_{vv}^+ - 2\ell_{uv}^+ \right). \tag{3}$$

(b) If $\ell_{ij}^{-1,[r]}$ denotes the entry $ij$ of the inverse of the matrix $L^{[r]}$ (the Laplacian $L$ after removing the row and the column corresponding to node $r$), then

$$w(\mathcal{F}_u^v) = \begin{cases} w(\mathcal{T}) \left( \ell_{uu}^{-1,[r]} + \ell_{vv}^{-1,[r]} - 2\ell_{uv}^{-1,[r]} \right) & \text{if } r \neq u, v \\ w(\mathcal{T}) \ell_{uu}^{-1,[v]} & \text{if } r = v \text{ and } u \neq v \\ w(\mathcal{T}) \ell_{vv}^{-1,[u]} & \text{if } r = u \text{ and } u \neq v. \end{cases} \tag{4}$$

*Proof.* We apply the matrix tree theorem (Theorem $A.1$) in combination with Lemma $A.2$, Lemma $A.3$ and Lemma $A.4$. We will use the following notation

$$\tilde{L}_G = L_G + \frac{\mathbb{1}\mathbb{1}^\top}{|V|}.$$

In order to use Lemma $A.4$ we will use the edge $\bar{e} = \{u, v\}$ with $w(\bar{e}) = 1$. Moreover, let us denote $b_{\bar{e}} = \mathbb{1}_u - \mathbb{1}_v$ where $\mathbb{1}_v$ indicates the column $v$ of the identity matrix. Since the difference between $G$ and $G_{\bar{e}}$ is just the edge $\bar{e}$ with $w(\bar{e}) = 1$, we can write the following relation between the Laplacians of $G$ and $G_{\bar{e}}$

$$L_{G_{\bar{e}}} = L_G + b_{\bar{e}} b_{\bar{e}}^\top.$$

Therefore, we may write

$$w(\mathcal{F}_u^v) \underbrace{=}_{\text{Lemma } A.4} w(\mathcal{T}_{G_{\bar{e}}}) - w(\mathcal{T}_G) \underbrace{=}_{\substack{\text{MTT} \\ \text{Theorem } A.1}} \frac{1}{|V|} \det \Big( \underbrace{\tilde{L}_G + b_{\bar{e}} b_{\bar{e}}^\top}_{\tilde{L}_{G_{\bar{e}}}} \Big) - \frac{1}{|V|} \det(\tilde{L}_G)$$

$$\underbrace{=}_{\text{Lemma } A.2} \frac{1}{|V|} \det(\tilde{L}_G) \left( 1 + b_{\bar{e}}^\top \tilde{L}_G^{-1} b_{\bar{e}} \right) - \frac{1}{|V|} \det(\tilde{L}_G) = \frac{1}{|V|} \det(\tilde{L}_G) \left( b_{\bar{e}}^\top \tilde{L}_G^{-1} b_{\bar{e}} \right)$$

$$\underbrace{=}_{\text{Lemma } A.3} \frac{1}{|V|} \det(\tilde{L}_G) \left( b_{\bar{e}}^\top \left( L^+ + \frac{\mathbb{1}\mathbb{1}^\top}{|V|} \right) b_{\bar{e}} \right) = \frac{1}{|V|} \det(\tilde{L}_G) \left( \ell_{uu}^+ + \ell_{vv}^+ - 2\ell_{uv}^+ \right)$$

$$\underbrace{=}_{\substack{\text{MTT} \\ \text{Theorem } A.1}} w(\mathcal{T}_G) \left( \ell_{uu}^+ + \ell_{vv}^+ - 2\ell_{uv}^+ \right)$$

The second statement can be deduced from equation (3) in the main paper. Still we show how it can be computed by following a similar argument as in the previous case. Let $r \neq u, v$.

$$w(\mathcal{F}_u^v) \underbrace{=}_{\text{Lemma } A.4} w(\mathcal{T}_{G_{\bar{e}}}) - w(\mathcal{T}_G) \underbrace{=}_{\substack{\textbf{MTT} \\ \text{Theorem } A.1}} \det\left(\underbrace{L_G^{[r]} + b_e^{[r]}\left(b_e^{[r]}\right)^\top}_{L_{G_{\bar{e}}}^{[r]}}\right) - \det(L_G^{[r]})$$

$$\underbrace{=}_{\text{Lemma } A.2} \det(L_G^{[r]})\left(1 + \left(b_{\bar{e}}^{[r]}\right)^\top \left(L_G^{[r]}\right)^{-1} b_{\bar{e}}^{[r]}\right) - \det(L_G^{[r]})$$

$$= \det(L_G^{[r]})\left(\ell_{uu}^{-1,[r]} + \ell_{vv}^{-1,[r]} - 2\ell_{uv}^{-1,[r]}\right) = w(\mathcal{T}_G)\left(\ell_{uu}^{-1,[r]} + \ell_{vv}^{-1,[r]} - 2\ell_{uv}^{-1,[r]}\right).$$

For $r = u$ the proof is the following:

$$w(\mathcal{F}_u^v) \underbrace{=}_{\text{Lemma } A.4} w(\mathcal{T}_{G_{\bar{e}}}) - w(\mathcal{T}_G) \underbrace{=}_{\substack{\textbf{MTT} \\ \text{Theorem } A.1}} \det\left(\underbrace{L_G^{[u]} + \mathbf{1}_v^{[u]}\left(\mathbf{1}_v^{[u]}\right)^\top}_{L_{G_e}^{[u]}}\right) - \det(L_G^{[u]})$$

$$\underbrace{=}_{\text{Lemma } A.2} \det(L_G^{[u]})\left(1 + \left(\mathbf{1}_v^{[u]}\right)^\top \left(L_G^{[u]}\right)^{-1} \mathbf{1}_v^{[u]}\right) - \det(L_G^{[u]}) = \det(L_G^{[u]})\ell_{vv}^{-1,[u]}$$

$$= w(\mathcal{T}_G)\ell_{vv}^{-1,[u]}.$$

The case $r = v$ is analogous. $\qquad\square$

## B   Proof of Theorem 4.1

**Theorem B.1** (Theorem 4.1)**.** The probability $x_q^{s_1}$ that a random walker as defined in [6] starting at node $q$ reaches $s_1$ first before reaching $s_2$ is equal to the Probabilistic Watershed probability defined in Definition 3.1 of the main paper:
$$x_q^{s_1} = P(q \sim s_1).$$

*Proof.* If we write the probability in terms of the inverse of $L^{[s_2]}$ (Lemma 2.2, equation 2 of the main paper) we find:

$$P(q \sim s_1) = \left(w(\mathcal{F}_{s_2}^q) + w(\mathcal{F}_{s_1}^{s_2}) - w(\mathcal{F}_{s_1}^q)\right) / \left(2w(\mathcal{F}_{s_1}^{s_2})\right)$$
$$= \left(\ell_{qq}^{-1,[s_2]} + \ell_{s_1 s_1}^{-1,[s_2]} - \left(\ell_{s_1 s_1}^{-1,[s_2]} + \ell_{qq}^{-1,[s_2]} - 2\ell_{qs_1}^{-1,[s_2]}\right)\right) / \left(2\ell_{s_1 s_1}^{-1,[s_2]}\right) = \ell_{qs_1}^{-1,[s_2]} / \ell_{s_1 s_1}^{-1,[s_2]}. \quad (5)$$

Therefore, to calculate the probabilities for $P(q \sim s_1)$ for every $q$ we only need to compute the column $s_1$ of $\left(L^{[s_2]}\right)^{-1}$. Solving the following linear system:

$$L^{[s_2]}y = \mathbf{1}_{s_1}/\ell_{s_1 s_1}^{-1,[s_2]} \iff y = \left(L^{[s_2]}\right)^{-1}\mathbf{1}_{s_1}/\ell_{s_1 s_1}^{-1,[s_2]} = \left(L^{[s_2]}\right)^{-1}_{\cdot, s_1}/\ell_{s_1 s_1}^{-1,[s_2]}, \quad (6)$$

where $\mathbf{1}_u$ denotes the column $u$ of the identity matrix, we have that $y$ is the vector formed by the elements in the right hand side of (5). Let us assume without loss of generality that the row corresponding to the seed $s_1$ is the first one, then we can express equation (6) block-wise :

$$\begin{pmatrix} L_{s_1 s_1} & B_{s_1}^\top \\ B_{s_1} & L_U \end{pmatrix}\begin{pmatrix} y_{s_1} \\ y_U \end{pmatrix} = \begin{pmatrix} L_{s_1 s_1}y_{s_1} + B_{s_1}^\top y_U \\ B_{s_1}y_{s_1} + L_U y_U \end{pmatrix} = \begin{pmatrix} 1/\ell_{s_1 s_1}^{-1,[s_2]} \\ 0 \end{pmatrix}, \quad (7)$$

where $L_{s_1 s_1}$ is the entry $s_1 s_1$ of the Laplacian $L^{[s_2]}$, $B_{s_1}$ is the row $s_1$ of this Laplacian without considering the element in the diagonal and $L_U$ are the rows and columns of the unseeded vertices. Since $y_{s_1} = P(s_1 \sim s_1) = 1$, we obtain the following linear system of equations

$$L_U y_U = -B_{s_1},$$

which is the same linear system that the Random Walker solves ([6] section III.B, equation (10)). Therefore $P(q \sim s_1) = y_q = x_q^{s_1}$ for all $q$. $\qquad\square$

## C Power Watershed

In this section we recall some definitions of [3]. Let $G = (V, E, w)$ be an undirected edge-weighted graph and $s_1, s_2 \in V$ two seeds as it has been considered in the main paper. In [3] the following objective function is proposed:

$$x^* = \arg\min_x \sum_{e=(v,u)\in E} (w(e))^\alpha \left( |x_v - x_u| \right)^\beta, \text{ s.t. } x_{s_1} = 1, \ x_{s_2} = 0. \tag{8}$$

This objective generalizes a set of segmentation algorithms depending on the choice of parameters $\alpha$ and $\beta$. For instance, $\alpha = 1$ and $\beta = 2$ give the Random Walker's objective function.

The Power Watershed algorithm solves (8) when $\alpha \to \infty$. The algorithm is similar to Kruskal's algorithm [8]: a maximum weight spanning forest rooted in the seeds is computed iteratively, but at each plateau (maximal connected subgraphs with constant edge-weight) the following optimization problem is solved

$$\min_x \sum_{(u,v)\in E} |x_u - x_v|^\beta. \tag{9}$$

In case that $\beta = 2$ this is equivalent to apply the Random Walker on the plateau.

## D Proof of Theorem 5.1

**Theorem D.1** (Theorem 5.1). Given two seeds $s_1$ and $s_2$, let us denote the potential of node $q$ being assigned to seed $s_1$ by the Power Watershed with $\beta = 2$ as $x_q^{\text{PW}}$. Let further $w_{\max}$ be $\max_{f \in \mathcal{F}_{s_1}^{s_2}} w(f)$. Then

$$x_q^{\text{PW}} = \frac{\left| \{ f \in \mathcal{F}_{s_1,q}^{s_2} \ : \ w(f) = w_{\max} \} \right|}{\left| \{ f \in \mathcal{F}_{s_1}^{s_2} \ : \ w(f) = w_{\max} \} \right|} =: P_\infty(q \sim s_1).$$

*Proof.* It has already been proven in Theorem 3 of [3] that the potential computed by the algorithm of the Power Watershed is equal to the limit of the Random Walker probabilities when the weights are raised to $\alpha \to \infty$. Since the Probabilistic Watershed probabilities are the same as the Random Walker probabilities (see Section 4), we just need to show that the limit of the Probabilistic Watershed with the weights raised to $\alpha \to \infty$ is counting MSFs.

$$P_\alpha(q \sim s_1) := \frac{\sum\limits_{f\in\mathcal{F}_{s_1,q}^{s_2}} \prod\limits_{e\in f} w(e)^\alpha}{\sum\limits_{f\in\mathcal{F}_{s_1}^{s_2}} \prod\limits_{e\in f} w(e)^\alpha} = \frac{\sum\limits_{f\in\mathcal{F}_{s_1,q}^{s_2}} w(f)^\alpha}{\sum\limits_{f\in\mathcal{F}_{s_1}^{s_2}} w(f)^\alpha} = \frac{\sum\limits_{f\in\mathcal{F}_{s_1,q}^{s_2}} \left( \dfrac{w(f)}{w_{\max}} \right)^\alpha}{\sum\limits_{f\in\mathcal{F}_{s_1}^{s_2}} \left( \dfrac{w(f)}{w_{\max}} \right)^\alpha} \xrightarrow[(\star)]{\alpha\to\infty} P_\infty(q \sim s_1).$$

$$(10)$$

In $(\star)$ we used the fact that $\frac{w(f)}{w_{\max}} < 1 \iff w(f) \neq w_{\max}$. When $\alpha \to \infty$, only for the MSFs the fraction $(w(f)/w_{\max})^\alpha$ does not tend to 0, but to 1. Thus, we are counting MSFs. $\qquad\square$

## E Effect of the entropy on the Probabilistic Watershed

Figure 1 illustrates how the forest distribution's entropy interpolates between (Power) Watershed and Probabilistic Watershed / Random Walker with decreasing sensitivity to edge-costs.

## F Edge and node probabilities in the Power Watershed

In this chapter, we elaborate the minimum spanning forest (mSF) counting interpretation of the Power Watershed. Figure 2 shows a graph $G$ with a single plateau $P$, a maximal connected subgraph of constant edge-cost $c$. To simplify our exposition, we made sure that there is exactly one path with maximum cost below $c$ from each seed to $P$. The nodes at the end of these paths are called $p_1$ and $p_2$,

Figure 1: Effect of the inverse temperature $\mu$ on Probabilistic Watershed solutions. (1a) shows a graph with 4 seeds and edge costs, $c(e)$. All paths from the query node $q$ to a seed $s_i$ have the same cost (only indicated once per seed). (1b) - (1d) show the Probabilistic Watershed's segmentation for edge weights $\exp(-\mu\,c(e))$. As $\mu$ grows, $q$'s assignment changes from a weight-independent (maximum entropy) one over two Random Walker assignments to the Watershed assignment (lowest entropy).

respectively. We illustrate the mSF-counting nature of the Power Watershed both on nodes and on edges.

In Figure 2a, we show the probability of an edge being present in a mSF. Outside the plateau, the edges are either part of every or of no mSF. All mSFs agree on these edges. They can be found by a variant of Kruskal's greedy algorithm which iteratively adds edges of minimal cost, while avoiding cycles and connections between the two seeds. Therefore, the edges outside the plateau are only black or white in Figure 2a. On the plateau all spanning forests have the same, minimal cost. Here, Power Watershed performs the Random Walker, or - in our forest-framework - counts spanning forests. Therefore, the edges on $P$ typically have a probability of being present in a mSF strictly between $0$ and $1$. Note that the final segmentation can be read-off from the edge probabilities in Figure 2a outside the plateau (as in each mSF in our example every node outside the plateau can be reached from a seed without entering the plateau) but not on the plateau without the node potentials.

In Figures 2b-2d, we show how likely an edge is connected to either of the seeds in a mSF. Again, all the edges outside the plateau are either always connected to the same seed in all mSFs or never part of any mSF. In the latter case, the conditional probability in 2b is not defined; we colored them white, which corresponds to the uninformed probability of $0.5$ for ease of presentation. The closer an edge of $P$ is to the node $p_1$, where the subtree of $s_1$ connects to the plateau, the higher its probability to be connected to $s_1$ among the mSFs that contain this edge (Figure 2b) and also among all mSFs (Figure 2c). The same holds for $s_2$ in Figures 2b and 2d. Note that in both Figure 2c and Figure 2d the color intensity of every edge $e = \{u, v\}$ is at most as high as that of $u$ or $v$. This is because whenever $e$ is connected to some seed in a mSF $f$, both $u$ and $v$ are connected to that seed in $f$, too.

We computed the probability of an edge being present in a spanning forest on the plateau by the generalization of the MTT in Lemma 1.9 of [5], see also Theorem 2 of [10] for a version on unweighted graphs. Then for each edge $e = \{u, v\}$ on $P$, we merged $u$ and $v$ into a new node $q_e$, thus obtaining a minor $P_e$ of $P$. On $P_e$ in turn, we computed the Probabilistic Watershed probabilities $P_{P_e}(p_1 \sim q_e)$, hence finding the share of 2-forests in $P_e$ isolating $p_1$ and $p_2$ that connect $q_e$ to $p_1$. This is nothing but the share of 2-forests in $P$ separating $p_1$ and $p_2$ that contain $e$ and connect it to $p_1$ among the 2-forests separating $p_1$ and $p_2$ that contain $e$. Multiplying this with the probability that an edge is part of any mSF gives the share of mSFs in $G$, which contain $e$ and connected it to $s_1$, among all mSFs separating $s_1$ and $s_2$.

(a) $P(\text{node} \sim s_1)$ and
$P(\text{edge} \in \text{some mSF})$

(b) $P(\text{node} \sim s_1)$ and
$P(\text{edge} \sim s_1 | \text{edge} \in \text{some msF})$

(c) $P(\text{node} \sim s_1)$ and
$P(\text{edge} \sim s_1, \text{edge} \in \text{some mSF})$

(d) $P(\text{node} \sim s_2)$ and
$P(\text{edge} \sim s_2, \text{edge} \in \text{some mSF})$

Figure 2: Power Watershed result on a grid graph with seeds $s_1$, $s_2$ and with random edge-costs outside a plateau of edges with the same cost (wide edges). By the results in Theorem 5.1, the Power Watershed counts mSFs. This is illustrated with both the node- and edge-colors. (2a-2d) The nodes are colored by their probability of belonging to seed $s_1$ ($s_2$), i.e. by the share of mSFs that connect a given node to $s_1$ ($s_2$). (2a) The edge-color indicates the share of mSFs in which the edge is present. (2b) The edge-color indicates the share of mSFs in which the edge is connected to seed $s_1$ among the mSFs that contain the edge. (2c - 2d) The edge-color indicates the share of mSFs in which the edge is connected to $s_1$ or $s_2$, respectively, among all mSFs.

## G  Rough lower bound for the number of forests in a grid graph

In this chapter we derive a rough lower bound on the number of spanning forests that separate $k$ given seeds in a two-dimensional grid graph. We refer to these forests as "$k$-forests". If there is some $n \times m$ subgrid without any seeds then the number of $k$-forests is at least as large as the number of spanning trees in the subgrid. This is because there are $k$-forests in which all nodes in the subgrid belong the tree of some of the seeds. We can compute the number of spanning trees $N_T$ in a grid graph with $n$ rows and $m$ columns by the closed-form formula (see [12] Theorem 1):

$$N_T(n, m) = \frac{2^{nm-1}}{nm} \cdot \prod_{\substack{i=0,\dots,n-1, \\ j=0,\dots,m-1, \\ (i,j)\neq(0,0)}} \left( 2 - \cos\left(\frac{i\pi}{n}\right) - \cos\left(\frac{j\pi}{m}\right) \right) \tag{11}$$

The image in Figure 2 of the main paper has a seed-free part of size $87 \times 272$, see Figure 3 below. This yields the following lower bound for the number of 13-forests separating the 13 seeds:

$$N_T(87, 272) \approx 10^{11847} \tag{12}$$

Figure 3: The shaded region was used to obtain a rough lower bound on the number of forests separating the seeds. There are about $10^{11847}$ spanning trees in the grid graph that corresponds to the shaded region and hence at least as many forests in the whole graph which separate the seeds.

## Footnotes

[1] Given a family of sets $\{A_i : i \in I\}$ the disjoint union is defined as $\bigsqcup_{i \in I} A_i = \bigcup_{i \in I} \{(x, i) : x \in A_i\}$.