[Reviews · NeurIPS 2019]

Reviewer 1



--- Summary --- The authors derive the Probabilistic Watershed: Given an undirected graph with seed nodes, the Probabilistic Watershed is a tractable method for computing marginals over *all seed-separating spanning forests* of nodes being assigned to seeds. The authors prove that the probabilities they obatin are equivalent to the probabilities yielded by the Random Walker algorithm. The authors state that this result has been shown in the original Random Walker work, yet is little known, and their proof is different and more self-contained, not relying on potential theory. Excitingly, their way of proof yields a novel interpretation of the Random Walker / Probabilistic Watershed probabilities in terms of the triangle equation on effective resistances between graph nodes. Last but not least the authors relate their theory to the Power Watershed, again yielding an exciting new insight, namely that for parameters beta=2 and alpha towards infinity, the latter computes marginals over all seed-separating *maximum* spanning forests (i.e. counts). --- Comments --- The paper is very clearly written and structured. The math is sound as far as I can see, and the notation is clear and convenient to read. The novel insights on Random Walker and Power Watershed are fundamental in that they offer concise and intuitive interpretations in both cases. I found this paper a pleasure to read. --- Minor Comments --- Typo line 128 denotes --> denote Line 147 "given its entropy" --> Hard to read (what is "it"?) Line 172 ff: For better readability, please introduce this paragraph by a short motivation. (in particular, it would be convenient to see in advance if it can be skipped to read on in line 182) Line 185: Again, it would ease readability here if you could shortly motivate what comes next. --- Post Author Feedback The examples for practical impact are very interesting; It would be great if you could add them to the paper or supplement.

Reviewer 2



Watershed is a well known and well studied segmentation procedure in graph theory. Over the last 20 years or so, our understanding of graph-based segmentation procedures have improved tremendously. This paper is an excellent contribution with a detailed stochastic approach that provides new interpretation of some links between watershed and other well-known graph-based procedures. Another key contribution is the uncertainty evaluation of the provided segmentation. Finally It shows how probabilistic watershed can provide better results by sampling over a larger set of forests.

Reviewer 3



The paper's is of theoretical nature and contains no empirical results or evidence to support the effectiveness of the proposed probabilistic watershed algorithm. On the theoretical side, its contribution seems to be limited to indicate equivalence, or bring new interpretation, to exisiting algorithms.

[Author Response · NeurIPS 2019]

We appreciate the time invested in carefully reading our paper and the very helpful and detailed comments by all reviewers. We will address their concerns in the revised version and thank the reviewers for pointing out that: our work is very clearly written, structured (R1) and detailed (R2), with a clear and convenient notation and a sound math (R1), and provides new links between and new interpretations of well-known graph-based procedures (R1, R2, R3).

The numbered citations will refer to the references of the paper.

**Reproducibility (R2):** The Random Walker and Watershed algorithms are well-known and implemented, e.g., in scikit-image's `segmentation.random_walker` and vigra's `analysis.watersheds`. Based on R2's comment, we will make the code and the edge-weights used for our illustrative computations available on GitHub for easy reproducibility.

**Connection to machine learning (R2):** In lines 11ff of the paper we state that seeded segmentation is essentially the same as graph-based semi-supervised learning, thus presenting our work as an instance of transductive learning. Given a limited number of labelled observations, the categories of the unlabelled data are inferred from their relation to the labelled data. We also cite machine learning literature that uses the Random Walker algorithm (and thus our Probabilistic Watershed) under different names in lines 49f [5, 40, 41] or combines it with deep leaning [9, 37]. We are happy to elaborate these connections further in the revised version of the paper.

**Empirical support (R3)**: Indeed the focus of our paper is conceptual and theoretical, as pointed out in the introduction (line 45). We have rigorously proven the equivalence of the Probabilistic Watershed to the Random Walker (theorem 4.1) and that a limit case calculates the potentials of the Power Watershed (theorem 5.1). The empirical effectiveness of the Random Walker and the Power Watershed algorithm *and thus of the equivalent Probabilistic Watershed* has been thoroughly established when these algorithms were introduced [13, 23, 41] and when they were subsequently applied. For instance, the Random Walker has been recently applied in the references below and in [9, 37]. In view of this vast empirical support, we decided not to add further experiments but to concentrate on deepening the theoretical insight in (connections between) different graph-based seeded segmentation algorithms, thus placing our contribution in the "Theory" subject area of NeurIPS 2019.

**Potential practical implications (R1 + R3)**: Although practical implications were not the focus of our work, the improved understanding of the aforementioned graph-based seeded segmentation algorithms and the introduced techniques may lead to new practical algorithms. An example of a new point of view provided by our work is the entropy regularised spanning forest sampling. It explains the interpolation between a Voronoi tessellation, the Random Walker and the Watershed algorithm discussed in [13] by varying the temperature of the Gibbs-distribution over all spanning forests. We have included an illustration of the temperature's effect in Figure 1, which we will also add to the appendix of the paper. Another idea inspired by our work would be to apply the Probabilistic Watershed framework to directed graphs, where one would sample directed spanning forests with the seeds as sinks to segment the unlabelled nodes. This might lead to a new practical algorithm for semi-supervised learning on directed graphs such as social/citation or Web networks and could be related to directed random walks.

(a) Graph with four seeds    (b) $\mu = 0$    (c) $\mu = 10$    (d) $\mu = 100$    (e) $\mu \to \infty$, Watershed

Figure 1: Effect of the inverse temperature $\mu$ on Probabilistic Watershed solutions. 1a shows a graph with 4 seeds and edge costs, $c(e)$. All paths from the query node $q$ to a seed $s_i$ have the same cost, (only indicated once per seed). 1b - 1d show the Probabilistic Watershed's segmentation for edge weights $\exp(-\mu\,c(e))$. As $\mu$ grows, $q$'s assignment changes from a weight-independent one over two Random Walker assignments to the Watershed assignment.

# References

Bockelmann, N. et al. (2019). Sparse annotations with random walks for U-Net segmentation of biodegradable bone implants in synchrotron microtomograms. In *International Conference on Medical Imaging with Deep Learning – Extended Abstract Track*.

Bui, V. et al. (2018). An automatic random walk based method for 3d segmentation of the heart in cardiac computed tomography images. In *ISBI*.

Liu, Z. et al. (2018). A method for PET-CT lung cancer segmentation based on improved random walk. In *ICPR*.


[Meta-Review · NeurIPS 2019]

Novel interpretation for the probabilistic watershed making to connection to the random walker algorithm. Brings new insights to old algorithms accept.